# Expanding global vaccine manufacturing capacity: Strategic prioritization in small countries

Sanjana Mukherjee[1]*, Kanika Kalra[1], Alexandra L. Phelan[2,3]

1 Center for Global Health Science and Security, Department of Microbiology and Immunology, Georgetown University, Washington, District of Columbia, United States of America, 2 Department of Environmental Health and Engineering, Johns Hopkins Bloomberg School of Public Health, Johns Hopkins University, Baltimore, Maryland, United States of America, 3 Center for Health Security, Johns Hopkins Bloomberg School of Public Health, Johns Hopkins University, Baltimore, Maryland, United States of America

* sm3783@georgetown.edu

**Data Availability Statement:** All relevant data are within the manuscript and Supporting Information files.

**Funding:** This work was funded by the Carnegie Corporation of New York (GR-21-58414) (ALP). The funders had no role in study design, data

## Abstract

The COVID-19 pandemic highlighted significant gaps in equitable access to essential medical countermeasures such as vaccines. Manufacturing capacity for pandemic vaccines, therapeutics, and diagnostics is concentrated in too few countries. One of the major hurdles to equitable vaccine distribution was "vaccine nationalism", countries hoarded vaccines to vaccinate their own populations first which significantly reduced global vaccine supply, leaving significant parts of the world vulnerable to the virus. As part of equitably building global capacity, one proposal to potentially counter vaccine nationalism is to identify small population countries with vaccine manufacturing capacity, as these countries could fulfill their domestic obligations quickly, and then contribute to global vaccine supplies. This cross-sectional study is the first to assesses global vaccine manufacturing capacity and identifies countries with small populations, in each WHO region, with the capacity and capability to manufacture vaccines using various manufacturing platforms. Twelve countries were identified to have both small populations and vaccine manufacturing capacity. 75% of these countries were in the European region; none were identified in the African Region and South-East Asia Region. Six countries have facilities producing subunit vaccines, a platform where existing facilities can be repurposed for COVID-19 vaccine production, while three countries have facilities to produce COVID-19 mRNA vaccines. Although this study identified candidate countries to serve as key vaccine manufacturing hubs for future health emergencies, regional representation is severely limited. Current negotiations to draft a Pandemic Treaty present a unique opportunity to address vaccine nationalism by building regional capacities in small population countries for vaccine research, development, and manufacturing.

## Introduction

Manufacturing capacity for pandemic vaccines, therapeutics, and diagnostics is concentrated in too few countries. During the COVID-19 pandemic, safe and effective vaccines have been

collection and analysis, decision to publish, or preparation of the manuscript.

**Competing interests:** The authors have declared that no competing interests exist.

developed at a record-breaking speed. However, global accessibility and affordability of these vaccinations has been unjustly inadequate. Eight of every ten vaccination doses produced has gone to high income countries [1]. As of June 2022, more than 5·22 billion people worldwide have received at least one dose of vaccine for COVID-19, accounting for 66·3% of the world population [2]. However, these numbers mask the inequity when disaggregated by region or income status. Only 18% of people in low-income countries have received at least one dose, compared to 80% of people in high income countries. The disparity is even more stark when it comes to fully vaccinated populations or populations that have received booster doses. Vaccine procurement and manufacturing has been especially difficult for low- and middle-income countries (LMICs) [1]. This has been partly due to the scale of vaccine nationalism seen throughout the pandemic: in particular high-income countries using legal and policy measures, such as advance purchase agreements and export controls, to divert or prevent the global equitable distribution of vaccines. Global governance efforts to allocate global vaccine supply on public-health-need have not been successful in rectifying these disparities: the World Health Organization's Strategy to Achieve Global COVID-19 Vaccination missed its goal of vaccinating 70% of the world's population by mid-2022 [2], and while the COVAX facility has resulted in more than 1 billion doses shipped to over 144 countries [3], it was also vulnerable to acts of vaccine nationalism, including the Indian government halting export of vaccines, leaving the COVAX Facility without vaccines, and the global supply constraints resulting from high income countries such as US, the European Union (EU) and Canada buying up supply [4–6].

This situation is enabled by the fact that global vaccine manufacturing capacity remains largely in high income countries, who have benefited from imperialism and colonialism to have the resources for manufacturing capacities. This issue extends well beyond COVID-19. According to the 2022 WHO Vaccine Market Report, globally, only 10 manufacturers provide 70% of non-COVID-19 vaccine doses [7]. An estimated 55% of vaccine manufacturing capacity is in East Asia and 40% in Europe and North America. This leaves Africa and South America with less than 5% of worldwide vaccine manufacturing capacity [8]. Furthermore, current intellectual property laws magnify inequities in access to life-saving and essential medical products, discriminating against low-income economies resulting in poor health outcomes [9].

Such inequity demands reform. Global health institutions have sought to address this inequity in a variety of ways, including through WTO Members adoption of a TRIPS waiver [10], and in proposed text for technology transfer in the draft Pandemic Treaty [11], but reform must include equitably increasing global vaccine manufacturing capacity. We have a rare opportunity to "reflect, reimagine and reset the world in a more just manner" [12].

Given the urgency of the task, there are a range of strategies that have been proposed to prioritize increasing global manufacturing capacities. Regional equity is a critical consideration: ensuring appropriate geographic distribution of manufacturing supply. For instance, WHO is leading efforts to create an mRNA vaccine technology transfer hub in South Africa to increase mRNA vaccine production in LMICs [13]. One additional strategy is the prioritization of building capacity in small population countries. The goal of this strategy is to directly limit vaccine nationalism by easily fulfilling national vaccine need well below full manufacturing capacity, removing the incentive to constrain global or regional supply. Thus, this study aims to assess overall global vaccine manufacturing capacity and identify small population countries with the capacity to manufacture vaccines, potentially contributing to global vaccine supplies in future health emergencies. Furthermore, we present a thorough overview of the existing global vaccine manufacturing capacity. Although previous studies and databases have collated data on countries and companies possessing vaccine manufacturing capabilities [14–16], our

study is pioneering in its inclusion of information pertaining to vaccine technology/platform, the various stages of vaccine production, and the history of WHO pre-qualification.

## Methods

### Study design

We designed a cross-sectional study to assess global vaccine manufacturing capacity as of March 1, 2022. Furthermore, we identified countries with small populations, defined as less than 15 million people, which had the capacity and capability for vaccine manufacturing, coding for different stages of the vaccine development and manufacturing pipeline (Fig 1). We then identified countries with very small populations, defined as less than 5 million people, regardless of existing vaccine manufacturing capacity as potential sites for vaccine manufacturing capacity to be established. Since this study does not involve clinical data or human participants, it was not submitted for ethics review.

### Data collection and extraction

We conducted an online search on country vaccine manufacturing capacity including on COVID-19 vaccine development and production activity. Information was extracted from: 1) academic publications and reports, 2) websites and reports from international organizations and non-governmental organizations, 3) websites and reports from private pharmaceutical companies, academic or public institutions with vaccine manufacturing and distribution experience and, 4) local and international news media reports. The detailed methodology is provided in S1 Text and the list of variables collected for each country is available in S1 Table.

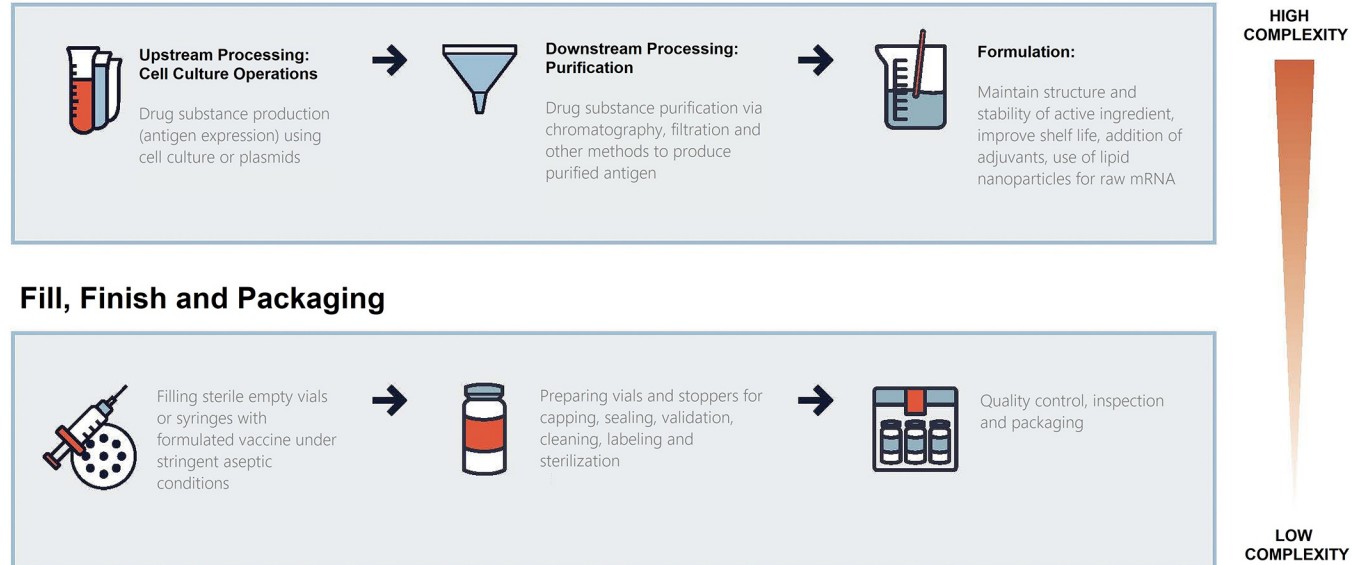

**Fig 1. Generic steps in vaccine development and production.** Vaccine manufacturing processes are highly complex and differ based on the type of vaccine manufacturing platform used for production. The upstream processing, downstream processing and formulation steps vary significantly based on biologic-derived vaccine manufacturing processes (inactivated, live-attenuated, subunit and viral vector vaccines) and chemical-based RNA vaccine (mRNA) manufacturing processes. Fill, finish, packaging, quality assurance and quality control steps, however, occur in the same manner across different vaccine manufacturing platforms. Information for generating figure adapted from the literature [8, 17].

## Data analysis

Data were managed using Airtable (Airtable, San Francisco, CA, USA) and MS Excel. 'Small countries' were defined as countries with a population less than 15 million, whereas 'very small countries' were defined as countries with a population of less than 5 million. A country was defined as having the capacity for vaccine production if it contained at least one documented manufacturing facility with prior/current vaccine production activity identified in our search (S1 Data).

We also identified countries with WHO prequalified vaccines. WHO's vaccines prequalification is used by United Nation agencies, such as UNICEF, for the procurement of life-saving vaccines for immunization programs for a range of diseases, including influenza (seasonal and pandemic). By prequalifying vaccines, WHO applies "international standards to assess the safety, efficacy and quality of vaccines produced" [18]. To this end, WHO conducts regular manufacturing facility site inspections for good manufacturing practice (GMP). Additionally, vaccine prequalification also depends on close collaboration with a country's National Regulatory Authority (NRA) considered "functional"- defined as having been WHO-listed as operating at a minimum of maturity level 3 [19]. NRAs conduct regulatory oversight of WHO prequalified vaccines by conducting facility inspections, evaluating vaccine clinical performance manufactured in the country and vaccine post-marketing surveillance. Manufacturers can apply for "vaccine prequalification" only if the country NRA is "functional". While COVID-19 vaccines are not currently on WHO's list of priority products for UN prequalification, we use this variable as a proxy to identify small population countries that can produce vaccines for global COVID-19 and other future pandemic threats immunization programs. Since these countries have functional NRAs and have manufacturing sites following GMP, COVID-19 vaccines produced in these countries can potentially be used for procurement by UN agencies and other organizations.

All statistical analyses were carried out using MS Excel. ArcGIS Pro version 2·9 (Esri Inc, Redlands, CA, USA) was used for data visualization and the World Countries map package (Source: Esri Data and Maps) [20] was used as the base map.

## Results

### Small population countries with vaccine manufacturing capacity

Our analysis identified a total of 43 countries with vaccine manufacturing capacity (S2 Table); 5 were in the Eastern Mediterranean Region, 17 in the European Region, 3 in the African Region, 7 in the Region of the Americas, 7 in the Western Pacific Region and 4 in the South-East Asia Region. However, when stratified by population size, only 12 countries with small populations were identified to have vaccine manufacturing capacity (Fig 2). Of these 12 countries, 75% (9/12) are in the European Region and one each in the Region of Americas, Western Pacific Region, and Eastern Mediterranean Region. We did not identify any small population countries with vaccine manufacturing capacity in the African Region and South-East Asia Region.

The 12 countries identified in our analysis are Austria, Azerbaijan, Belgium, Bulgaria, Cuba, Czech Republic, Denmark, Serbia, Singapore, Sweden, Switzerland, and Tunisia. Vaccines manufactured in these countries vary widely, with 6 countries having portfolios of producing vaccines for 3 or more diseases (Belgium, Bulgaria, Cuba, Denmark, Serbia, and Sweden) (S2 Table). Currently, COVID-19 vaccines are manufactured in 4 countries with small populations (Belgium, Cuba, Czech Republic, and Switzerland) (Table 1); a COVID-19 vaccine candidate manufactured in Austria was terminated. Two COVID-19 vaccines are currently manufactured in Belgium, including Pfizer-BioNTech BNT162b2 COVID-19 Vaccine and Oxford/AstraZeneca (ChAdOx1-S [recombinant]) COVID-19 vaccine. Belgium has both

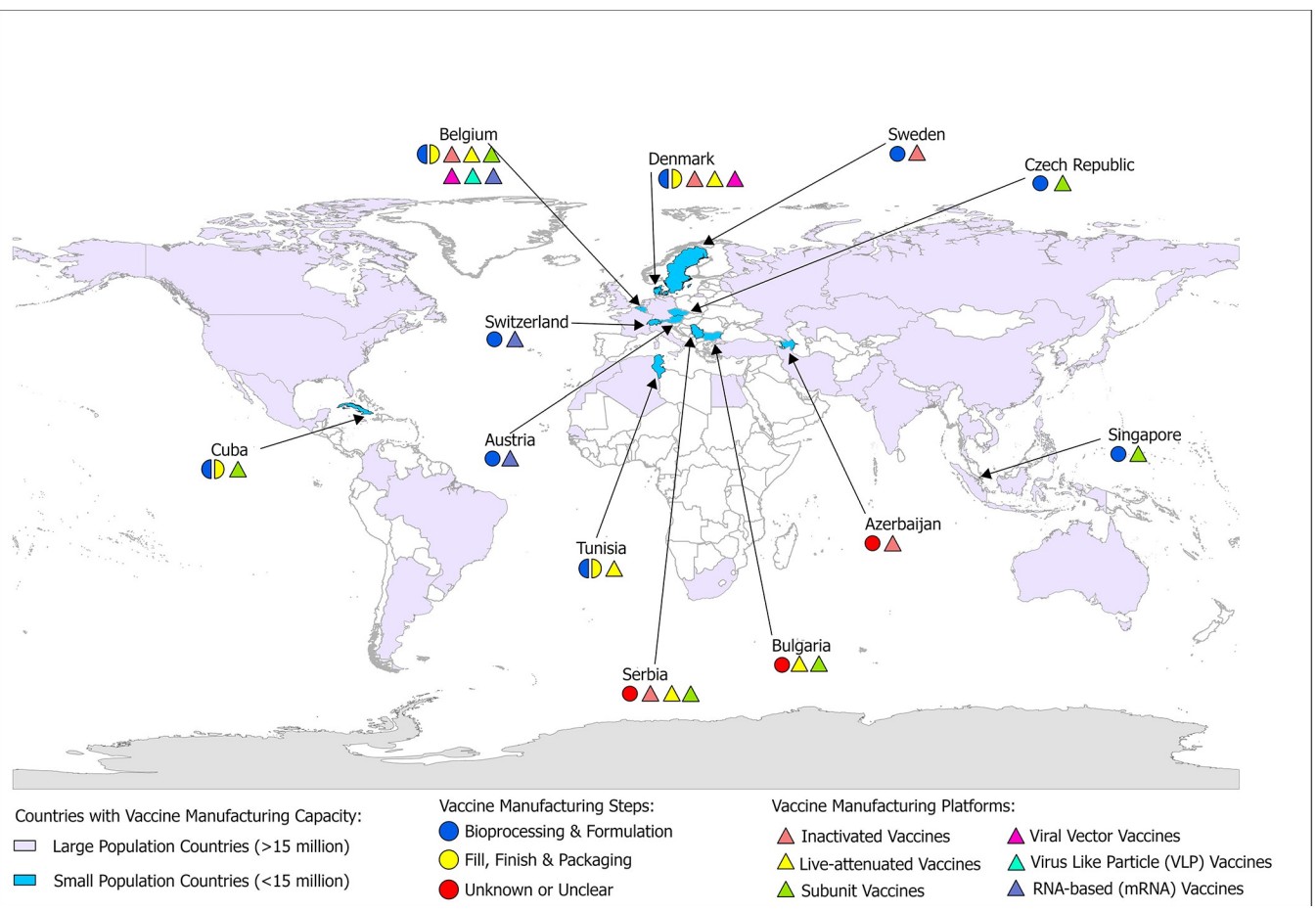

**Fig 2. Countries with small populations (< 15 million) identified to have vaccine manufacturing capacity as of March 1, 2022.** Small population countries with vaccine manufacturing capacity (n = 12) are highlighted in blue, while those with populations >15 million are highlighted in green. Countries with unknown or absent vaccine manufacturing capacity are shaded gray. The figure also highlights a country's capacity to manufacture vaccines using various vaccine manufacturing platforms (inactivated vaccines, live-attenuated vaccines, subunit vaccines, virus-like particle vaccines, viral-vector vaccines, and RNA based vaccines) and the vaccine manufacturing steps present ('Bioprocessing and formulation' capacity, 'Fill, finish, and packaging', 'Unknown/Unclear'). World Countries map package (Source: Esri Data and Maps) was used as the base map [20].

'Bioprocessing & Formulation' and 'Fill, Finish & Packaging' capacity for manufacturing Pfizer-BioNTech COVID-19 vaccines. Additionally, manufacturing facilities in Belgium also carry out 'Bioprocessing & Formulation' of Oxford/AstraZeneca COVID-19 vaccines. Czech Republic and Switzerland, both in the European Region, also have facilities manufacturing COVID-19 vaccines. Cuba, present in the Region of the Americas, manufactures and produces two protein subunit COVID-19 vaccines: Soberana 2 FINLAY-FR-2 vaccine and Abdala CIGB-66 vaccine. Since both manufacturers of these Cuban vaccines operate in a "closed cycle" platform, these facilities have the capacity and capability to carry out the complete development cycle of a vaccine from research and development to production and marketing [21]. Our assessment did not identify any small population countries manufacturing COVID-19 vaccines in the other four WHO regions.

## Vaccine manufacturing platforms in small population countries

To identify countries with the potential to repurpose existing vaccine manufacturing platforms and installed vaccine manufacturing bases for expansion of COVID-19 vaccine production,

**Table 1. COVID-19 vaccine production in countries with small populations as of March 1, 2022.** For each country, the type of COVID-19 vaccine produced, the names of manufacturing facilities, steps of vaccine production and vaccine manufacturing platform are included.

| Country | WHO Region | COVID-19 Vaccine | Manufacturing Facility Name and Location | COVID-19 Vaccine Production Step | Vaccine Platform |
|---|---|---|---|---|---|
| Belgium | European Region | Pfizer-BioNTech BNT162b2 vaccine | Pfizer (Puurs, Belgium) | Bioprocessing and Formulation Fill, Finish and Packaging | RNA based (mRNA) |
| | | Oxford/AstraZeneca (ChAdOx1-S [recombinant]) vaccine | Novasep (Seneffe, Belgium) | Bioprocessing and Formulation | Viral Vector Vaccine |
| Cuba | Region of the Americas | Soberana 2 FINLAY-FR-2 vaccine | The Finlay Institute (Havana, Cuba) | Bioprocessing and Formulation Fill, Finish and Packaging | Subunit Vaccine |
| | | Abdala CIGB-66 vaccine | Center of Genetic Engineering and Biotechnology (CIGB) (Havana, Cuba) | Bioprocessing and Formulation Fill, Finish and Packaging | Subunit Vaccine |
| Czech Republic | European Region | Novavax NVX- CoV2373 vaccine | Praha Vaccines acquired by Novavax (Bohumil, Czech Republic) | Bioprocessing and Formulation | Subunit Vaccine |
| Switzerland | European Region | Moderna mRNA-1273 vaccine | Lonza (Visp, Switzerland) | Bioprocessing and Formulation | RNA based (mRNA) |
| | | Pfizer-BioNTech BNT162b2 vaccine | Novartis (Stein, Switzerland) | Fill, Finish and Packaging | RNA based (mRNA) |

we assembled a list of all countries with vaccine manufacturing capacities and the platforms used to produce vaccines for any infectious disease (S2 Table). Additionally, manufacturing capacity also depends on the vaccine manufacturer's capacity and capability to manufacture the drug substance or active pharmaceutical ingredient (Bioprocessing & Formulation) and provide fill-and-finish services (Fig 1). A full list of a country's "Bioprocessing & Formulation" and "Fill, Finish and Packaging" capacity is provided in S2 Table.

Overall, the 43 countries identified in this study had a wide range of vaccine manufacturing platforms; inactivated vaccine platforms were the most common (n = 26) followed by subunit vaccine platforms (n = 23), viral vector vaccine platforms (n = 21) and live-attenuated vaccines (n = 20). Eight countries were identified to have and/or potentially have the capacity to manufacture mRNA vaccines.

In the 12 small population countries, subunit vaccine platforms (n = 6), inactivated vaccines platforms (n = 5) and live-attenuated vaccine platforms (n = 5) are the most common (Fig 2). Additionally, 25% (3/12) of the small population countries have the capacity to manufacture/ potentially manufacture RNA based mRNA vaccines; these countries are Austria, Belgium, and Switzerland, all of which are present in the European Region. Belgium is the only country which has manufacturing facilities utilizing six vaccine manufacturing platforms. Manufacturers utilizing subunit vaccine platforms, the platform most commonly present in the small population countries identified in this study and which have the potential to be repurposed for COVID-19 production, are present in Belgium, Bulgaria, Cuba, Czech Republic, Serbia, and Singapore. Majority of the countries with subunit vaccine platforms are present in the European Region (4/6), along with one each in the Region of the Americas (Cuba) and the Western Pacific Region (Singapore). While Singapore produces active pharmaceutical ingredients for different vaccines using the subunit vaccine platform, it does not have fill-finish capacity and is currently not producing finished vaccines (S2 Table). Vaccine manufacturers utilizing viral vector vaccines are present in Belgium and Denmark, which is another potential platform that can be repurposed to expand COVID-19 vaccine production.

A list of COVID-19 vaccines, the vaccine manufacturing platforms used for their production and the countries producing these vaccines are provided in S3 Table. While most of these

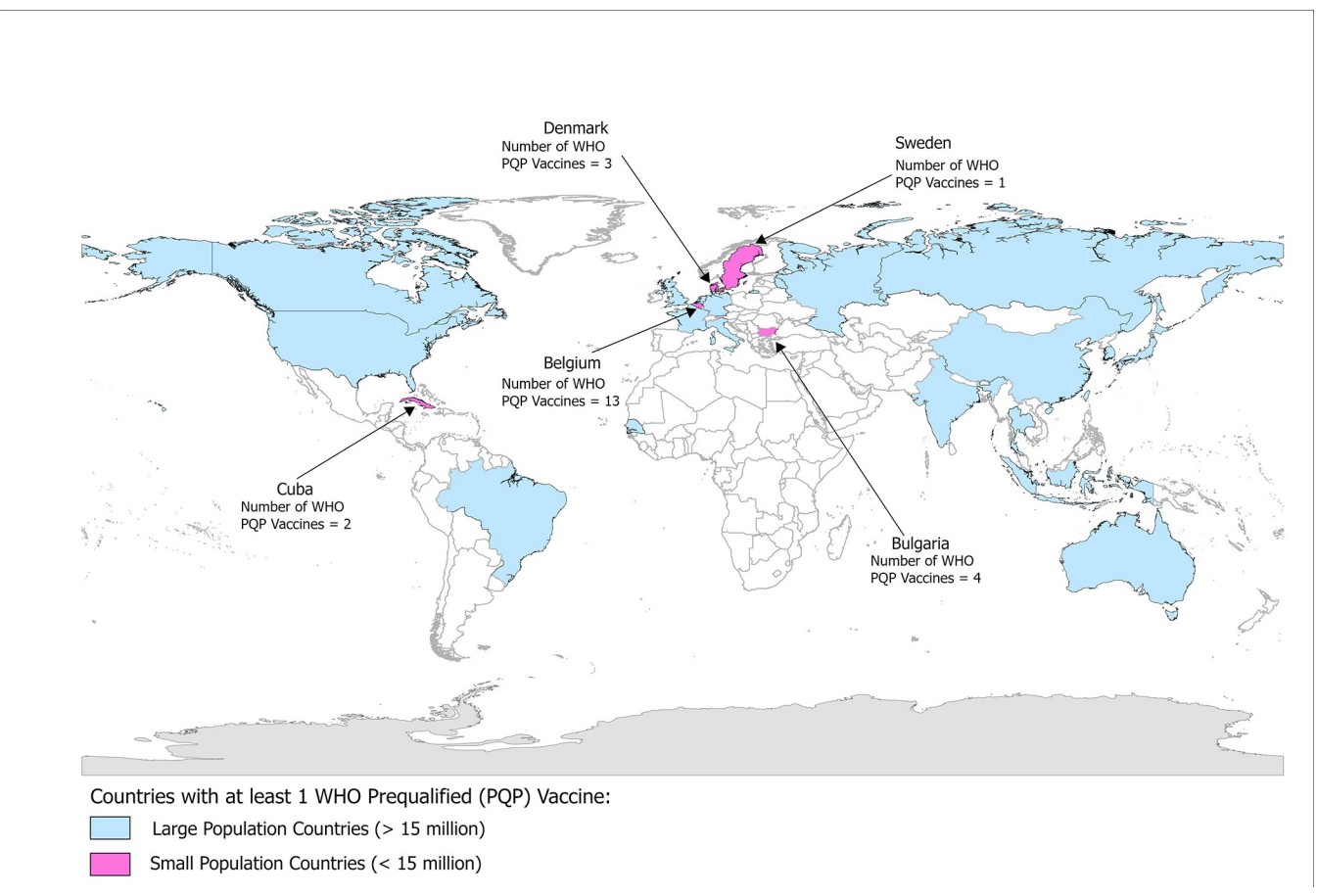

**Fig 3. Countries with small populations (< 15 million) identified to produce at least one WHO prequalified vaccine.** Small population countries with manufacturers producing a WHO prequalified vaccine (n = 5) are highlighted in pink, while those with populations >15 million are highlighted in blue. World Countries map package (Source: Esri Data and Maps) was used as the base map [20].

vaccines are RNA-based mRNA vaccines (n = 6), vaccines using subunit vaccine platforms (n = 4) and viral vector vaccine platforms (n = 4) are also common.

## Countries with WHO prequalified vaccines

Overall, only 22 countries have manufacturing facilities with prior/current WHO prequalified vaccines (Fig 3). 45·4% (n = 10) of these countries are present in the European Region, 18·2% (n = 4) are in the Region of the Americas, 18·2% (n = 4) are in the Western Pacific Region, and 13·6% (n = 3) are in the South-East Asia Region. Senegal is the only country in the African Region producing a prequalified vaccine (Yellow Fever vaccine). There are no countries in the Eastern Mediterranean Region producing WHO prequalified vaccines. Of the 22 countries, 5 small population countries have manufacturing facilities producing at least one prequalified vaccine—Belgium, Bulgaria, Cuba, Denmark, and Sweden (Fig 3). 80% of these countries (n = 4) countries are in the European region. None of these 5 countries produce WHO prequalified influenza vaccines.

## Very small population countries regardless of vaccine manufacturing capacity

To identify very small population countries where vaccine manufacturing capacity and drug regulatory system capacity can be built, we compiled a list of countries with populations less

than 5 million. While these countries may not currently have existing vaccine manufacturing or regulatory capacity, countries from this list can be identified to potentially serve as vaccine manufacturing hubs if resources and attention are directed to initiate the development of vaccine manufacturing facilities with GMPs and establishing medical product regulatory systems. By geographically diversifying vaccine manufacturing in such countries with very small populations, security, and stability to large parts of the world can be ensured during future health emergencies.

We identified 73 countries with populations less than 5 million (S1 Fig). None of these countries were identified to have existing vaccine manufacturing capacity in our analysis. When stratified by WHO region, 4 countries were in the Eastern Mediterranean Region, 20 countries in the European Region, 15 countries in the African Region, 15 countries in the Region of the Americas, 16 countries in the Western Pacific Region and 3 in the South-East Asia Region.

## Discussion

Preempting vaccine nationalism can take direct or indirect paths. The ongoing negotiations for a Pandemic Treaty is an opportunity to directly address vaccine nationalism through legally binding obligations. This may include limiting the use of advance purchase agreements (APAs) and export controls during a pandemic, such as prohibiting their use, or imposing maximum limits for both, based on real time production capacity or public health justifications. Prohibiting the use of APAs and export controls is unlikely to be politically palatable, domestically, or internationally, while imposing maximum limits alone is unlikely to be sufficient to minimize the use of export controls or APAs. In addition, to be effective, the latter, along with the imposition of maximums, would require reporting, verification, and adjudication processes. Such a compliance mechanism may be feasible under a new international legal instrument such as the Pandemic Treaty but will require championing from negotiating Member States.

The ongoing Pandemic Treaty negotiations are also an opportunity to indirectly address vaccine nationalism by establishing legally binding obligations on States Parties to build global capacities for vaccine research, development, and manufacturing capacities. These efforts should be supported by an equity-focused and strategically informed approach to minimize the impacts of vaccine nationalism (especially in the absence of direct obligations) that may persist even when regional capacities have been developed.

One possible strategy for the short to medium term is to not only identify countries with existing manufacturing capacities that can be developed, but to map that against countries with small populations to prioritize capacity building and pre negotiated commitments. The goal of this strategy is to minimize the potential impact of vaccine nationalism on regional supply: even if a country with dedicated diagnostic, vaccine, and therapeutic production were to fulfill its domestic need first, with a small population, it will be rapidly fulfilled with minimized impact on regional distribution.

In our results, 12 countries have met the criteria of having current vaccine manufacturing capacity and populations below 15 million. However, the number and types of vaccines manufactured by these countries vary widely. For instance, while manufacturers in Belgium are producing more than 10 vaccines using six different vaccine manufacturing platforms, Tunisia, present in the Eastern Mediterranean Region, currently has the capacity to only produce 'bacillus Calmette–Guérin' (BCG) vaccines at the Pasteur Institute of Tunis (IP Tunis). The manufacturing hubs identified in our study are in the European Region, Region of Americas, Western Pacific Region, and Eastern Mediterranean Region. Importantly, we did not identify

any small population countries with vaccine manufacturing hubs in the African Region and South-East Asia Region, highlighting the disparity in regional vaccine producers.

Since production capacity varies widely among vaccines and vaccine manufacturing platforms, the potential for capacity expansion is not spread evenly amongst global vaccine makers. RNA-based vaccines are a novel vaccine manufacturing platform and used almost exclusively to produce COVID-19 vaccines. In our study, we identified Belgium, Switzerland, and Austria as the only small population countries with capacity for production of mRNA vaccines, all three of which are in the European region. The Pfizer facility in Puurs, Belgium has the capacity to carry out both "Bioprocessing & Formulation" and "Fill, Finish and Packaging" of the Pfizer-BioNTech COVID-19 Vaccine and has the capacity to produce approximately 1·3 billion shots annually [22, 23]. Meanwhile, Novartis has the capacity to conduct "Fill, Finish and Packaging" of the Pfizer-BioNTech COVID-19 Vaccine at its Stein, Switzerland facility [24], and Lonza manufactures the active pharmaceutical ingredient for the Moderna mRNA-1273 vaccine at its manufacturing site in Visp, Switzerland [25]. Since RNA-based vaccine platforms are new, repurposing existing facilities may not be possible as manufacturing of the mRNA vaccine drug substance requires installation of commercial-grade mRNA and lipid nanoparticles (LNPs) GMP manufacturing capacity [8]. While industry stakeholders have estimated that vaccine manufacturing capacity building can take three-to-four years [8, 26], efforts have been made to successfully accelerate this timeline. For instance, Pfizer established mRNA manufacturing capacity in its Puurs, Belgium and Kalamazoo, USA facilities in an estimated 100 days which included building formulation laboratories and designing industrial processes for mRNA vaccine production [22]. Additionally, in June 2021, WHO announced that it was supporting the development of an mRNA technology transfer hub in South Africa in collaboration with Biovac, Afrigen Biologics and Vaccines, a network of universities and the Africa Centres for Disease Control and Prevention (CDC) [27]. Seven months after the establishment of the mRNA hub, in January 2022, Afrigen was reported to have produced the first batch of drug product formulation for an mRNA vaccine candidate [28]. Additionally, 15 companies are currently in the process of learning to make such mRNA vaccines at Afrigen [29]; two of these institutes were identified in our study as they are present in small population countries–Torlak Institute in Serbia and Pasteur Institut Tunis in Tunisia. A report also identified more than 100 manufacturers across Asia, Africa and Latin America with the capacity to manufacture mRNA vaccines based on the manufacturers' current capacity to manufacture sterile injectables [30]. The establishment of mRNA vaccine manufacturing hubs globally will allow for greater and diversified vaccine manufacturing capability resulting in strengthening of regional health security and responding more equitably to future pandemics.

Since many manufacturing facilities globally employ subunit and viral vector vaccine platforms for vaccine production, rapid and effective repurposing using existing installed facilities and bioreactors is possible. Indeed, estimates suggest that 1–5% of existing subunit and viral vector vaccine manufacturing capacity can be repurposed to expand the production of COVID-19 vaccines [8]. Additionally, protein subunit vaccines may have advantages over other types of vaccines including relatively inexpensive production, suitability for people with compromised immune systems, and stability at a broad range of temperatures making vaccine supply chain and logistics (vaccine storage, distribution, handling, and management) easier [31–33]. As shown in S3 Table, currently, at least three COVID-19 vaccines employ subunit vaccine platforms; Novavax NVX- CoV2373 protein-based vaccine is currently being manufactured in multiple countries. In our results, we identified six small population countries with the capacity to manufacture subunit vaccines—Belgium, Bulgaria, Cuba, Czech Republic, Serbia, and Singapore. Since these countries have existing subunit vaccine manufacturing capacities, facilities in these countries can be repurposed to produce subunit COVID-19 vaccines.

Indeed, to develop the Soberana 2 FINLAY-FR-2 vaccine, the Finlay Institute of Vaccines in Havana, Cuba repurposed its existing 'conjugate' vaccine technology to produce the Soberana 2 COVID-19 vaccine and has the capacity to produce 10 million Soberana 2 doses per month [31]. Additionally, we identified two small population countries (Belgium and Denmark) with the capacity to manufacture viral vector vaccines, which was identified as another platform with the potential to be repurposed for expanding COVID-19 vaccine production. We also identified numerous small population countries with inactivated and live-attenuated vaccine platforms; however, repurposing these facilities for COVID-19 vaccine production may not be possible due to specific viral containment requirements for handling live viruses [8].

A country's vaccine manufacturing capacity is also dependent on the steps of vaccine production available at facilities. While centralized production involves manufacturing vaccines and other health products at scale in centrally located sites, distributed manufacturing involves the production of different components of the vaccine at different geographical locations and facilities [34]. For example, while Lonza produces the active pharmaceutical ingredient for the Moderna mRNA-1273 vaccine in Visp, Switzerland, CordenPharma produces lipid nanoparticles required for the Moderna mRNA-1273 vaccine in Boulder, Colorado, USA and the final fill-finish steps are carried out by Catalent in Bloomington, Indiana, USA [25]. Such a scenario is also seen in Singapore, the only small population country in the Western Pacific Region with vaccine manufacturing capacity, where only active ingredients are manufactured and not finished vaccines [35]. However, efforts are being made to include Singapore as a producer of finished vaccines, to expand vaccine manufacturing capacity in the region. In 2020, Thermo Fisher Scientific Inc. announced plans to develop two new sterile filling lines in Singapore to expand Singapore's fill-finish capacity for vaccines and other products [36]. Improving a region's capacity to produce both the active ingredient for vaccines and finished vaccines is critical for responding to future pandemic threats.

Throughout the course of the COVID-19 pandemic, rapid scale up of manufacturing capacity has been observed to meet the global demand for COVID-19 vaccines. This included formation of more than 150 partnerships with contract development and manufacturing companies (CDMOs) and other vaccine manufacturers, globally [8]. Despite this, vaccine manufacturing and supply of COVID-19 vaccines remains disproportionate and inequitable. Based on our results, we have identified multiple small population countries with the capacity to manufacture different components of vaccines using various vaccine manufacturing platforms, although a majority of these are in the European region. An efficient strategy to increase COVID-19 vaccine supply is for innovator companies to enter agreements to transfer vaccine technology and knowledge to manufacturers in such small population countries.

We also identified five small population countries which have a history of manufacturing WHO prequalified vaccines, four of which are in the European Region (Belgium, Bulgaria, Denmark, and Sweden) and one in the Region of the Americas (Cuba). Vaccines manufactured in these countries can be used for procurement of vaccines by UN agencies for responding to future pandemic threats as these countries have "functional" NRAs and have manufacturing sites following GMP. The presence of a competent, effective, and transparent regulatory authority within a country is crucial to ensure access to safe and effective medical products to consumers and protect consumers from substandard or falsified medical products. Without the presence of functional NRAs, countries cannot assess whether vaccines and other health products being manufactured locally meet approved quality control standards [18, 37]. It is reported that, of the 194 Member States, only 26% (50 countries) contain mature regulatory agencies, while 144 countries have suboptimal regulatory systems [38]. Thus, in the short to medium term, expansion of COVID-19 vaccine production should prioritize manufacturers in small population countries with functional NRAs, while long-term goals to expand regional

vaccine manufacturing capacity should also include elements to strengthen regulatory capacity of countries.

In addition to the availability of vaccine manufacturing capacity in strategic regional hubs, it is crucial to bolster the vaccine input supply chain which includes raw materials necessary for production in all stages of vaccine manufacturing. Examples of such critical raw materials include syringes, glass vials, bioreactor bags, filters, and cell culture media. During the COVID-19 pandemic, numerous countries instituted export restrictions on such key raw materials, significantly disrupting global supply chains [39, 40]. The Pandemic Treaty is an opportunity for States to also address export bans, restrictions and other trade-related measures of crucial goods required for production of essential medical countermeasures and identify countries with small populations that can produce such essential raw materials during future pandemics.

To improve access and availability of drugs, vaccines, and other medical products during health emergencies, a comprehensive policy toolkit is required that employs both long-term and short-term strategies. Lessons learned from the COVID-19 pandemic have highlighted the need to establish and strengthen national and regional vaccine manufacturing capacity in LMICs, with many countries pledging to do so. For instance, in September 2022, the African Union called for a new for a 'New Public Health Order for Africa' to expand "manufacturing of vaccines, diagnostics, and therapeutics to democratize access to life-saving medicines and equipment" [41]. Improving the overall vaccine manufacturing ecosystem, including establishing vaccine manufacturing hubs and strengthening medical product regulatory systems, would likely fall under long-term strategies, given the significant time and investment required to build research & development infrastructure, train workforce, and create financing arrangements. Thus, in this study, we propose that small population countries with vaccine manufacturing capacity and functional NRAs can aid in global vaccine supply by supporting vaccine procurement for facilities like the COVAX facility. Furthermore, procurement commitments by UN agencies and export of vaccines can stimulate private sector investment in vaccine and therapeutics manufacturing in the small population countries identified in this study, aiding in the establishment of a financing mechanism to bolster in-country manufacturing efforts. However, as seen during the COVID-19 pandemic, vaccine export restrictions and vaccine nationalism was a major barrier to vaccine access. Indeed, during the pandemic, the EU placed COVID-19 vaccine export restrictions and required authorization for the export of vaccines outside the EU [6, 42]. This is particularly important, as four of the five small countries with both vaccine manufacturing capacities and functional NRAs identified in our study, are EU member states. To tackle this, policy proposals to limit the use of export restrictions during future health emergencies have been introduced, with the EU highlighting that vaccine producing countries "should be ready to export a fair share of their domestic production" [43]. Thus, the Pandemic Treaty provides a unique opportunity to negotiate such obligations from vaccine producing countries, including small vaccine manufacturing countries, to help improve vaccine access, globally.

## Conclusions

The inequitable distribution of lifesaving COVID-19 vaccines has resulted in vastly different COVID-19 responses and outcomes. While many high income and upper-middle income countries have vaccinated their populations against the SARS-CoV-2 virus, many resource-constrained countries have been unable to secure vaccines for their populations and are left vulnerable to the continuing devastating impact of this disease. A pandemic treaty which addresses the issue of inequitable access to essential medical countermeasures is crucial to

respond to future emerging infectious disease threats. By outlining legally binding obligations for State Parties to build global capacities for vaccine research, development, and manufacturing in strategic locations in all WHO regions, the treaty would provide an opportunity to ensure that future pandemic preparedness and response is centered on principles of equity and justice.

## Supporting information

**S1 Text. Methods.**
(DOCX)

**S1 Fig. Countries with very small populations (< 5 million).** While these countries may not have existing vaccine manufacturing capacity or regulatory systems, countries from this list can be identified to potentially serve as vaccine manufacturing hubs, while ensuring geographical diversity. World Countries map package (Source: Esri Data and Maps) was used as the basemap [20]. The countries are: Albania, Andorra, Antigua and Barbuda, Armenia, Bahamas, Bahrain, Barbados, Belize, Bhutan, Bosnia and Herzegovina, Botswana, Brunei Darussalam, Cabo Verde, Comoros, Cook Islands, Croatia, Cyprus, Djibouti, Dominica, Equatorial Guinea, Eritrea, Estonia, Eswatini, Fiji, Gabon, Gambia, Georgia, Grenada, Guinea-Bissau, Guyana, Iceland, Ireland, Jamaica, Kiribati, Kuwait, Latvia, Lesotho, Lithuania, Luxembourg, Maldives, Malta, Marshall Islands, Mauritania, Mauritius, Federated States of Micronesia, Monaco, Mongolia, Montenegro, Namibia, Nauru, New Zealand, Niue, North Macedonia, Palau, Panama, Qatar, Republic of Moldova, Saint Kitts and Nevis, Saint Lucia, Saint Vincent and the Grenadines, Samoa, San Marino, Sao Tome and Principle, Seychelles, Slovenia, Solomon Islands, Suriname, Timor-Leste, Tonga, Trinidad and Tobago, Tuvalu, Uruguay, Vanuatu.
(TIF)

**S1 Table. Detailed information on variables collected per country.**
(DOCX)

**S2 Table. Global vaccine manufacturing capacity as of March 1, 2022.** The list includes vaccine portfolios and vaccine manufacturing procedures in each country.
(DOCX)

**S3 Table. List of COVID-19 vaccines manufactured by the 43 countries identified in our study as of March 1, 2022.** The vaccine manufacturing platforms used for their production and the countries producing these vaccines are included in this list.
(DOCX)

**S1 Data. Raw dataset of countries and companies with vaccine manufaturing capacity identified in our study as of March 1, 2022.**
(XLSX)

## Author Contributions

**Conceptualization:** Alexandra L. Phelan.

**Data curation:** Sanjana Mukherjee, Alexandra L. Phelan.

**Formal analysis:** Sanjana Mukherjee, Alexandra L. Phelan.

**Funding acquisition:** Alexandra L. Phelan.

**Investigation:** Sanjana Mukherjee, Kanika Kalra.

**Methodology:** Sanjana Mukherjee, Alexandra L. Phelan.

**Project administration:** Alexandra L. Phelan.

**Resources:** Alexandra L. Phelan.

**Software:** Sanjana Mukherjee.

**Supervision:** Alexandra L. Phelan.

**Visualization:** Sanjana Mukherjee.

**Writing – original draft:** Sanjana Mukherjee, Kanika Kalra, Alexandra L. Phelan.

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
