## [Decision Letter · Decision Letter 0]

25 Apr 2023

PGPH-D-23-00097

Expanding Global Vaccine Manufacturing Capacity: Strategic Prioritization in Small Countries

Dear Dr. Mukherjee,

Thank you for submitting your manuscript to PLOS Global Public Health. After careful consideration, we feel that it has merit but does not fully meet PLOS Global Public Health’s publication criteria as it currently stands. Therefore, we invite you to submit a revised version of the manuscript that addresses the points raised during the review process.

We look forward to receiving your revised manuscript.

Kind regards,

Madhukar Pai, MD, PhD

Editor-In-Chief

Journal Requirements:

1. Please send a completed 'Competing Interests' statement, including any COIs declared by your co-authors. If you have no competing interests to declare, please state "The authors have declared that no competing interests exist". Otherwise please declare all competing interests beginning with the statement "I have read the journal's policy and the authors of this manuscript have the following competing interests:"

2. Please provide separate figure files in .tif or .eps format.

Reviewers' comments:

Reviewer's Responses to Questions

**Comments to the Author**

1. Does this manuscript meet PLOS Global Public Health’s publication criteria? Is the manuscript technically sound, and do the data support the conclusions? The manuscript must describe methodologically and ethically rigorous research with conclusions that are appropriately drawn based on the data presented.

Reviewer #1: Yes

Reviewer #2: Yes

2. Has the statistical analysis been performed appropriately and rigorously?

Reviewer #1: N/A

Reviewer #2: N/A

3. Have the authors made all data underlying the findings in their manuscript fully available (please refer to the Data Availability Statement at the start of the manuscript PDF file)?

Reviewer #1: Yes

Reviewer #2: Yes

4. Is the manuscript presented in an intelligible fashion and written in standard English?

Reviewer #1: Yes

Reviewer #2: Yes

5. Review Comments to the Author

Reviewer #1: The authors have painstakingly addressed the issue of vaccine accessibility disparity that had existed between the first world and others for a long time which was made more visible during the COVID-19 pandemic. The design and the methodical approach at identifying low population countries with capacity and capability that could be supported technically to manufacture vaccines either singly or in collaboration with other entities seem to be reasonable to reduce the vaccine nationalism that was clearly demonstrated during the COVID pandemic. The need for strengthening these manufacturing facilities and their corresponding regulatory authorities for prequalification purposes were also emphasized. Africa has been in the eye of the storm and has been disproportionally affected during this period. With a huge population, poor per capita income and low capacity for production of molecular-derived vaccines e.g. mRNA, attention should be centred on how this region and allied regions could be assisted. Overall, this manuscript has highlighted the weaknesses of the strong vaccine producing nations to equitably protect the weaker ones during pandemic and has put up a lot of salient recommendations and suggestions on how the future pandemic could be approached based on the current experience.

Reviewer #2: General remarks:

This is an interesting paper and contribution to the lit on policy proposals for address equity in the next pandemic. Congratulations to the authors, and my apologies for the week’s delay in my review – I was very ill.

This is a good draft, but there remain some points in the argument that could be strengthened.

The introduction effectively reviews the failure of global vaccine distribution. The methods/results show where existing capacity is, with more detailed analysis for small/very small countries and prequal capacity. The argument that manufacturing capacity could be concentrated in small countries to mitigate vaccine nationalism is argued/theoretical.

The main weakness in the paper is that it does not effectively address counterarguments or limitations. The idea of concentrating manufacturing capacity in small countries is novel /contrary to the accepted wisdom that investments in manufacturing capacity should prioritize a) low income countries, b) countries in regions with less capacity, and c) countries that are large enough to maintain markets in non-pandemic times. The countries identified by the authors as the best candidates would rank low across all of these criteria. It’s great that the authors are injecting new ideas into this space, but give how far the findings are (ie including small, rich European countries as priority candidates), it should engage more with the existing lit / policy discussions and pre-empt counterarguments.

Some questions I had in reading the draft:

-How might excess capacity be financed in small countries? How would this capacity be used in not-pandemic times?

-I see the argument for small countries exhausting domestic demand, but domestic foreign policy / trade policy is still very much on the table. We can’t assume that the exhaustion of demand would lead countries to be willing to work with international efforts / market mechanisms.

-The elephant in the room is trade agreements/blocs/sanctions. At the center of the Venn diagram of prioritized countries (small, manufacturing capacity), history of prequal) are 5 countries: 4 in the European Union, and 1 under trade sanctions in much of the world. The former would still be subject to EU-level export controls until EU-wide demand was fulfilled, and the latter would be (and indeed, during the pandemic was) hampered in manufacturing and distribution but trade sanctions. This also links to financing… if we’re arguing for more-than-national capacity, one would expect some level of investment (private or public), and prioritizing rich European countries that did not deliver in the last pandemic is, for me, a bit of a hard sell.

Contribution to the lit- sell your results! Is this the first compilation of global vaccine manufacturing capacity by country? Say so! If not, explain how it fits in with existing databases / papers.

Relation to the lit – worth briefly outlining the lit on current proposals to improve equity, and how yours is complementary (or not).

Textual comments

Introduction:

64: More than just Canada! The authors argue that vaccine nationalism was a key factor here. This is generally accepted, but you might consider adding a box briefly summarizing/citing a few cases so Canada seems less exceptional

65-7: This argument would be strengthened by making explicit reference/explanation to key mechanisms here – for example, the global intellectual property regime.

Methods:

87 -93: I think this is possibly missing the last step. You identified capacity, identified small countries, and then….

103-10: Please specify in the Appendix search terms, databases used, dates and languages search, which pharmaceutical companies and international organizations, how news websites were searched etc. It doesn’t necessarily need to be a full formalized PRISMA-style scoping review but it needs to have enough basic information to be replicable.

111-112: I got confused because above it says small countries (5) and then very small (15), but here just small. Remind the reader of both.

112: One documented manufacturing facility, as identified from the lit review above? Or a single source/database?

Results:

127: “with vaccine manufacturing hubs” – it’s just any vaccine manufacturing right? Not hubs per se?

Fig 2 / methods: 6/12 countries identified are in the European Union, where export controls were (to my knowledge) largely at the EU and not sub-national level. It would be good to address why (or why not) you are still including small countries within the EU.

Nice S1 Appendix table.

164-65: Is this a typo? S2 looks likes it is assembled by company, not country.

S3: Why are some products in parentheses but not others in “manufacturing countries”?

196-201: This is almost word for word / the sentences are only very lightly rearranged from the WHO page. I noticed because I looked up the WHO page to double check something. Cite / add direct quotes where appropriate.

203: cite

196-212: Not really results – perhaps should be put further up and integrated within methods as this is essentially a subgroup analysis.

Discussion:

246-7: confusing wording “imposing maximums for both based on real time production capacity or requiring public health justification”. Imposing maximum export controls? As in number of controls?

247-251: As a reader I’m losing track of latter… is former/latter APAs/export controls, or is it prohibition/limits?

254: The ongoing Pandemic Treaty negotiations (worth specifying, as there are quite a few salient policy processes right now)

6. PLOS authors have the option to publish the peer review history of their article (what does this mean?). If published, this will include your full peer review and any attached files.

**Do you want your identity to be public for this peer review?** For information about this choice, including consent withdrawal, please see our Privacy Policy.

Reviewer #1: No

Reviewer #2: No

---

## [Editor Report · Decision Letter 1]

1 Jun 2023

Expanding Global Vaccine Manufacturing Capacity: Strategic Prioritization in Small Countries

PGPH-D-23-00097R1

Dear Dr. Mukherjee,

We are pleased to inform you that your manuscript 'Expanding Global Vaccine Manufacturing Capacity: Strategic Prioritization in Small Countries' has been provisionally accepted for publication in PLOS Global Public Health.

Best regards,

Madhukar Pai, MD, PhD

Editor-In-Chief
